applied mathematics/mathematical modelling/ inorganic chemistry

clock reaction, matched asymptotic expansions, mathematical chemistry

**Author for correspondence:**
D. J. Smith
e-mail: d.j.smith@bham.ac.uk

# Mathematical modelling of the vitamin C clock reaction

## R. Kerr, W. M. Thomson and D. J. Smith

School of Mathematics, University of Birmingham, Edgbaston, Birmingham B15 2TT, UK

 WMT, 0000-0001-9087-7495; DJS, 0000-0002-3427-0936

Chemical clock reactions are characterized by a relatively long induction period followed by a rapid 'switchover' during which the concentration of a *clock chemical* rises rapidly. In addition to their interest in chemistry education, these reactions are relevant to industrial and biochemical applications. A substrate-depletive, non-autocatalytic clock reaction involving household chemicals (vitamin C, iodine, hydrogen peroxide and starch) is modelled mathematically via a system of nonlinear ordinary differential equations. Following dimensional analysis, the model is analysed in the phase plane and via matched asymptotic expansions. Asymptotic approximations are found to agree closely with numerical solutions in the appropriate time regions. Asymptotic analysis also yields an approximate formula for the dependence of switchover time on initial concentrations and the rate of the slow reaction. This formula is tested via 'kitchen sink chemistry' experiments, and is found to enable a good fit to experimental series varying in initial concentrations of both iodine and vitamin C. The vitamin C clock reaction provides an accessible model system for mathematical chemistry.

## 1. Introduction

'Clock reactions' encompass many different chemical processes in which, following mixing of the reactants, a long induction period of repeatable duration occurs, followed by a rapid visible change. These reactions have been studied for over a century, with early examples including the work of H. Landolt on the sulphite–iodate reaction in the 1880s—for review, see Horváth & Nagypál [1], and the work of G. Forbes *et al.* [2] on G. Vortmann's thiosulphate–arsenite/arsenate reaction [3]. The origin of the term 'clock' is unclear, however Richards and Loomis in 1927 [4] referred to '…*the long familiar iodine "clock" depending upon the reduction of potassium iodate by sulfurous acid*…', suggesting that the term had already been in common use for some time.

Conway's 1940 article [5] referred to both reactions as common tools in the classroom for teaching the Law of Mass Action. An early example of detailed mathematical modelling of multi-step reactions via systems of nonlinear differential equations was

given by Chien [6]; this work exhibited a number of analytical techniques including solving a Riccati-type equation resulting from quadratic reaction kinetics (a step we will find useful in what follows). Anderson [7] demonstrated the computer solution of chemical kinetics problems, including the iodine clock reaction. Further review and history of the field can be found in [1,8].

Through dynamical systems analysis and the method of matched asymptotic expansions, Billingham & Needham [8] identified and studied two clock reaction mechanisms, both alone and in combination. The first mechanism is *autocatalysis*: the reaction producing the clock chemical is catalysed by the clock chemical itself, for example, the reaction $P + 2B \rightarrow 3B$, where $B$ is the clock chemical and $P$ is a precursor. This type of reaction is characterized by nonlinear kinetics; in the above case, the reaction rate for production of $B$ would be proportional to $pb^2$, where lower case letters correspond to chemical concentrations.

The second mechanism identified was *inhibition*. An inhibitor chemical $C$ removes the clock chemical, e.g. $B + C \rightarrow D$, keeping the concentration of $B$ low. Simultaneously, clock chemical is produced upstream from a precursor, e.g. via the reaction $P \rightarrow B$. Provided that the supply of precursor is sufficiently large relative to the initial concentration of inhibitor, the inhibitor will eventually be depleted, allowing the clock chemical concentration to rise. Billingham & Needham [8] then applied this framework to the iodate–arsenous acid system, characterizing the process as involving a combination of inhibition ($B + C \rightarrow 2A$) of the clock chemical and *indirect autocatalysis* in one of the reactants $A$. The catalysis is indirect because it is through the combination of this reaction with ($P + 5A \rightarrow 3B$) results in $A$ effectively catalysing its own production ($5A$ produce $6A$). For the iodate–arsenous acid system, the clock chemical $B$ is iodine ($I_2$), the inhibitor $C$ is arsenite ion $AsO_3^{3-}$, the precursor $P$ is iodate ($IO_3^-$) and the other reactant $A$ is the iodide ion ($I^-$). Further mathematical development can be found in [9,10], application of this approach to systems of industrial importance [11] and biochemistry [12]. Further discussion on the appropriateness of the term 'clock reaction' for various processes involving induction periods can be found in [13].

The system we will consider was described by Wright [14,15] as a variation on the iodine clock reaction requiring only safe household chemicals. A combination of iodine ($B$) and iodide ions ($A$) are supplied initially. In the presence of hydrogen peroxide, iodide ions are converted to iodine molecules,

$$2H^+(aq) + 2I^-(aq) + H_2O_2(aq) \rightarrow I_2(aq) + 2H_2O(l). \tag{1.1}$$

The rate of this reaction has been considered by a number of authors [16–18], who have observed that the rate is linear in both $I^-$ and $H_2O_2$ concentrations due to a rate-limiting step involving nucleophilic attack of $I^-$ on $H_2O_2$. However, these studies have worked with hydrogen peroxide concentrations which are similar to the other substrates, whereas we will work with hydrogen peroxide in great excess. In consequence, the $H_2O_2$ concentration may be omitted from the model, and a one-step reaction with rate that is quadratic in $I^-$ concentration (as would be expected from applying the law of mass action) is found to match well to experimental results. We therefore have in simplified form,

$$2A \rightarrow B, \quad \text{rate} \quad k_0 a^2. \tag{1.2}$$

Starch in solution appears blue in the presence of iodine. There is no separate precursor $P$ in this reaction.

The inhibitor $C$ is ascorbic acid—i.e. vitamin C—which converts iodine to iodide (alongside producing equal quantities of $H^+$ ions) via the reaction,

$$I_2(aq) + C_6H_8O_6(aq) \rightarrow 2H^+(aq) + 2I^-(aq) + C_6H_6O_6(aq) \tag{1.3}$$

or in simplified form,

$$B + C \rightarrow 2A, \quad \text{rate} \quad k_1 bc. \tag{1.4}$$

In contrast to the iodate–arsenous acid system, the pair of reactions (1.2) and (1.4) are not overall autocatalytic in either $A$ or $B$; the sum $a + 2b$ is conserved. The reaction has similar form to the reaction of iodide and peroxydisulfate in the presence of thiosulphate, which has been characterized by Horváth & Nagypál [1] as a *pure substrate-depletive* clock reaction (and therefore fitting Lente et al.'s strict definition of a clock reaction [13]). When sufficient inhibitor $C$ is present, the reaction (1.4) dominates, and iodine is held at relatively low concentration in comparison to iodide ions. However, the reaction (1.4) depletes vitamin C, and hence eventually this reaction can no longer continue. At this point, the slower reaction (1.2) dominates, resulting in concentration of the clock chemical iodine rising, and hence the solution turning from clear/white to blue. The time at which this reversal in the dynamics occurs is repeatable. A similar reaction involving persulfate in place of hydrogen peroxide has recently been analysed by Burgess & Davidson [19].

## 2. Ordinary differential equation model

From the law of mass action, the reactions (1.2) and (1.4) and the assumption that the system is well mixed and isothermal, lead to the ordinary differential equation model,

$$\frac{\mathrm{d}a}{\mathrm{d}t} = 2k_1 bc - 2k_0 a^2, \tag{2.1}$$

$$\frac{\mathrm{d}b}{\mathrm{d}t} = -k_1 bc + k_0 a^2 \tag{2.2}$$

and

$$\frac{\mathrm{d}c}{\mathrm{d}t} = -k_1 bc, \tag{2.3}$$

with initial conditions based on the initial concentrations of $A$, $B$ and $C$, respectively.

$$a(0) = a_0, \quad b(0) = b_0 \quad \text{and} \quad c(0) = c_0. \tag{2.4}$$

Inspecting equations (2.1) and (2.2), it is clear that the total concentration of iodine atoms $a(t) + 2b(t)$ is conserved. Denoting the initial concentration by $m_0 = a_0 + 2b_0$, we then have,

$$a(t) = m_0 - 2b(t), \tag{2.5}$$

which leads to the two-variable system,

$$\frac{\mathrm{d}b}{\mathrm{d}t} = -k_1 bc + k_0 (m_0 - 2b)^2 \tag{2.6}$$

and

$$\frac{\mathrm{d}c}{\mathrm{d}t} = -k_1 bc. \tag{2.7}$$

Non-dimensionalizing the system with the scalings,

$$b = m_0 \beta, \quad c = c_0 \gamma \quad \text{and} \quad t = (k_1 c_0)^{-1} \tau, \tag{2.8}$$

leads to the dimensionless initial-value problem,

$$\frac{\mathrm{d}\beta}{\mathrm{d}\tau} = -\beta\gamma + \epsilon\rho(1 - 2\beta)^2 \tag{2.9}$$

and

$$\frac{\mathrm{d}\gamma}{\mathrm{d}\tau} = -\rho\beta\gamma, \tag{2.10}$$

with initial conditions,

$$\beta(0) = \phi := \frac{b_0}{m_0} \quad \text{and} \quad \gamma(0) = 1, \tag{2.11}$$

where the dimensionless parameters $\rho = m_0/c_0$ and $\epsilon = k_0/k_1$. A key feature of the dynamics is that because the reaction producing clock chemical is much slower than the inhibitory reaction, the latter ratio is very small, i.e. $\epsilon \ll 1$. The ratio $\rho$ will be assumed to be order 1; it will also turn out to be that $\rho\phi := b_0/c_0 < 1$ in order for the vitamin C supply to survive the initial transient.

## 3. Qualitative analysis of the dynamics

Significant insight into the induction period of the system (2.9) and (2.10) can be obtained through an informal quasi-steady analysis, that is to exploit the fact that the clock chemical $B$ is slowly varying during this period. If $\mathrm{d}\beta/\mathrm{d}\tau \approx 0$ in equation (2.9), then it follows that $\beta\gamma \approx \epsilon\rho(1 - 2\beta)^2$. Substituting this expression into equation (2.10) for inhibitor depletion leads to,

$$\frac{\mathrm{d}\gamma}{\mathrm{d}\tau} \approx -\epsilon\rho^2(1 - 2\beta)^2. \tag{3.1}$$

Given that clock chemical concentration is small during the induction period, $\beta \approx 0$ and so equation (3.1) can be simplified as $\mathrm{d}\gamma/\mathrm{d}\tau \approx -\epsilon\rho^2$, therefore $\gamma \approx c_1 - \epsilon\rho^2\tau$, where $c_1$ is a constant which we will determine in §4.2. This expression shows that the length of the induction period scales with $(\epsilon\rho^2)^{-1}$ in dimensionless variables.

The system (2.9) and (2.10) has the unique equilibrium $(\beta, \gamma) = (1/2, 0)$ which represents the long-term fate of the system (zero inhibitor, all reactants converted to clock chemical). The eigenvalues of

the linearized system at this point are 0, with eigenvector $(1, 0)$ and $-\rho/2$, with eigenvector $(1, \rho)$. The zero eigenvalue indicates the slow manifold $\{(s, 0) : s \in \mathbb{R}\}$ which we will find the system approaches after the induction period is complete. The negative eigenvalue indicates a stable manifold $\{(s, \rho s) : s \in \mathbb{R}\}$ which lies outside the physically reasonable region of interest $0 \leq \beta \leq 1/2$, $0 \leq \gamma$.

# 4. Asymptotic analysis

The smallness of the reaction rate ratio $\epsilon$ enables an approximate solution to be sought via matched asymptotic expansions, in a similar manner to Billingham & Needham [8,9]. A visual summary of the asymptotic regions in the phase plane is shown in figure 1. We will assume throughout that the initial $I_2$ and vitamin C concentrations are such that $\rho\phi < 1$.

## 4.1. Initial adjustment (region I)

The first region that can be identified is where the independent variable $\tau$ and both dependent variables $\beta$, $\gamma$ are order 1. Seeking a solution of the form,

$$\beta = \beta_0 + \epsilon\beta_1 + \cdots \quad \text{and} \quad \gamma = \gamma_0 + \epsilon\gamma_1 + \cdots, \tag{4.1}$$

we find at leading order,

$$\frac{d\beta_0}{dt} = -\beta_0\gamma_0 \tag{4.2}$$

and

$$\frac{d\gamma_0}{dt} = -\rho\beta_0\gamma_0, \tag{4.3}$$

with initial conditions $\beta_0(0) = \phi$, $\gamma_0(0) = 1$. The quantity $\rho\beta_0(t) - \gamma_0(t)$ is therefore constant, leading to the separable ordinary differential equation,

$$\frac{d\beta_0}{d\tau} = -\beta_0(\rho(\beta_0 - \phi) + 1) \quad \text{and} \quad \beta_0(0) = \phi, \tag{4.4}$$

with solution,

$$\beta_0(\tau) = \frac{\phi(1 - \rho\phi)e^{(\rho\phi-1)\tau}}{1 - \rho\phi e^{(\rho\phi-1)\tau}}. \tag{4.5}$$

It follows that

$$\gamma_0(\tau) = \frac{\rho\phi(1 - \rho\phi)e^{(\rho\phi-1)\tau}}{1 - \rho\phi e^{(\rho\phi-1)\tau}} + 1 - \rho\phi. \tag{4.6}$$

As $\tau \to \infty$ we observe that $(\beta_0(\tau), \gamma_0(\tau)) \to (0, 1 - \rho\phi)$. This behaviour is the initial transient through which the system rapidly adjusts to quasi-equilibrium. During this interval, approximately $\rho\phi = b_0/c_0$ of the initial quantity of inhibitor is consumed. The next order terms in the dynamics may be found through straightforward but lengthy manipulations which do not reveal any significant further insight.

## 4.2. Induction (region II)

Once $\beta(\tau) = O(\epsilon)$ the system reaches a quasi-steady state. Rescaling $T = \epsilon\tau$ and $\hat{\beta}(T) = \epsilon^{-1}\beta(T/\epsilon)$, with $\hat{\gamma}(T) = \gamma(T/\epsilon)$, the system takes the form,

$$\epsilon\frac{d\hat{\beta}}{dT} = -\hat{\beta}\hat{\gamma} + \rho(1 - 2\hat{\beta})^2 \tag{4.7}$$

and

$$\frac{d\hat{\gamma}}{dT} = -\rho\hat{\beta}\hat{\gamma}. \tag{4.8}$$

Again seeking asymptotic expansions $\hat{\beta} = \hat{\beta}_0 + \epsilon\hat{\beta}_1 + \cdots$ and $\hat{\gamma} = \hat{\gamma}_0 + \epsilon\hat{\gamma}_1 + \cdots$ we have the leading order problem,

$$0 = -\rho\hat{\beta}_0\hat{\gamma}_0 + \rho \tag{4.9}$$

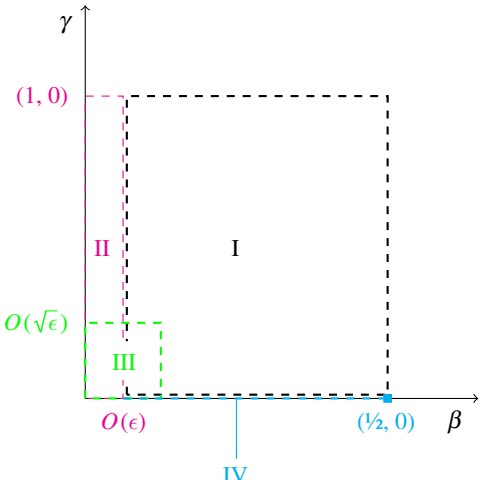

**Figure 1.** Schematic diagram of the asymptotic regions and equilibrium point (1/2, 0) in the $(\beta, \gamma)$–phase plane. If the system is started in region I, the time coordinate scales as $\tau = O(1)$ in region I, $\tau = O(\epsilon^{-1})$ in regions II and IV, and $\tau - \epsilon^{-1}[\rho^{-2} - \rho^{-1}\phi] = O(\epsilon^{-1/2})$ in region III.

and

$$\frac{d\hat{\gamma}_0}{dT} = -\rho\hat{\beta}_0\hat{\gamma}_0, \tag{4.10}$$

which has solutions of the form,

$$\hat{\gamma}_0 = c_1 - \rho^2 T \quad \text{and} \quad \hat{\beta}_0 = \frac{\rho}{1 - \rho\phi - \rho^2 T}. \tag{4.11}$$

Taking $T \to 0$ and matching to the region I solution as $\tau \to \infty$ shows that the constant $c_1 = 1 - \rho\phi$. In the original dimensionless variables, the leading order solution in region II is therefore,

$$\beta(\tau) = \frac{\epsilon\rho}{1 - \rho\phi - \rho^2\epsilon\tau} + O(\epsilon^2) \quad \text{and} \quad \gamma(\tau) = 1 - \rho\phi - \rho^2\epsilon\tau + O(\epsilon). \tag{4.12}$$

The solution (4.12) becomes non-uniform as $\tau \to (1 - \rho\phi)/(\rho^2\epsilon)$, which corresponds to the end of the induction period. In dimensional variables, the 'switchover' time is therefore characterized by,

$$t_{\text{sw}} = \frac{1 - b_0/c_0}{k_1 c_0 (m_0/c_0)^2 (k_0/k_1)} = \frac{c_0 - \phi m_0}{m_0^2 k_0}. \tag{4.13}$$

We will return to equation (4.13) to interpret the experimental results in §5.

## 4.3. Long-term state (region IV)

We now turn our attention to the asymptotic dynamics in the long-term state of the system after the switchover, in which $\beta(\tau)$ is again $O(1)$ and $\tau = O(\epsilon^{-1})$. The corner region III between II and IV will be addressed later. Denoting $\check{\beta}(T) = \beta(\epsilon^{-1}T)$ and $\check{\gamma}(T) = \gamma(\epsilon^{-1}T)$, the system takes the form,

$$\epsilon\frac{d\check{\beta}}{dT} = -\check{\beta}\check{\gamma} + \epsilon\rho(1 - 2\check{\beta})^2 \tag{4.14}$$

and

$$\epsilon\frac{d\check{\gamma}}{dT} = -\rho\check{\beta}\check{\gamma}. \tag{4.15}$$

Again substituting expansions of the form $\check{\beta} = \check{\beta}_0 + \epsilon\check{\beta}_1 + \cdots$ and $\check{\gamma} = \check{\gamma}_0 + \epsilon\check{\gamma}_1 + \cdots$ yields at leading order,

$$\check{\beta}_0\check{\gamma}_0 = 0, \tag{4.16}$$

hence $\check{\gamma}_0 = 0$, and moreover at $O(\epsilon^n)$,

$$\frac{d\check{\gamma}_{n-1}}{dT} = -\rho(\check{\beta}_0\check{\gamma}_n + \cdots + \check{\beta}_n\check{\gamma}_0). \tag{4.17}$$

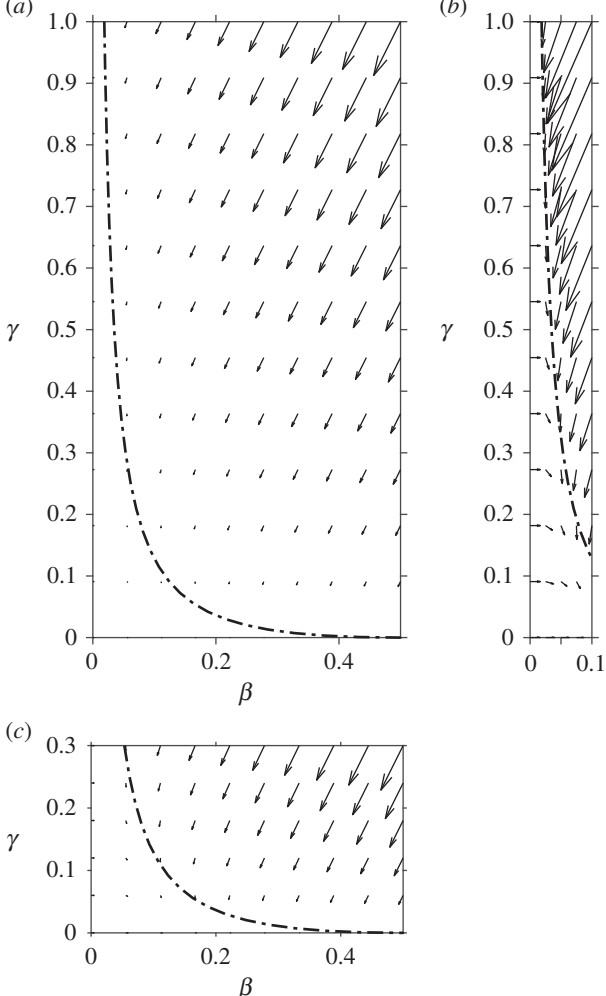

**Figure 2.** A numerical phase space diagram with phase directions and magnitudes shown by arrows and quasi-steady trajectory $\beta\gamma = \epsilon\rho(1 - 2\beta)^2$ by the dash-dot line; dimensionless groups $\epsilon = 0.01$ and $\rho = 2$. (b,c) Close-up detail for small $\beta$ and $\gamma$, respectively, with arrows rescaled for visibility.

Hence $\check{\gamma}_0 = \cdots = \check{\gamma}_{n-1} = 0$ implies that $\check{\gamma}_n = 0$. By induction it follows that $\check{\gamma}_n = 0$ for all $n$. Therefore, once $\beta = O(1)$ and $T = O(\epsilon^{-1})$, then $\gamma$ is zero at all orders in $\epsilon$. Taking $\check{\gamma} \approx 0$ then yields,

$$\frac{\mathrm{d}\check{\beta}}{\mathrm{d}T} = \rho(1 - 2\check{\beta})^2, \tag{4.18}$$

(note that we are now working with $\check{\beta}$ rather than just the leading order term) which has solution,

$$\check{\beta}(T) = \frac{1}{2} - \frac{1}{2(c_2 + 2\rho T)}, \tag{4.19}$$

$c_2$ being a constant. The matching condition $\beta(T) \to 0$ as $T \to \rho^{-2} - \rho^{-1}\phi$ yields $c_2 = 1 + 2\phi - 2\rho^{-1}$.

In the original variables,

$$\beta(\tau) \approx \frac{1}{2} - \frac{1}{2(1 + 2[\phi - \rho^{-1} + \rho\epsilon\tau])} \quad \text{and} \quad \gamma(\tau) \approx 0, \tag{4.20}$$

the error term in both solutions being beyond any $O(\epsilon^n)$.

## 4.4. Corner (region III)

It remains to match regions (II) and (IV), which occurs when $\beta$ is no longer $O(\epsilon)$ and $\gamma$ is no longer $O(1)$, i.e. in the bottom left corner of the phase space in figure 2. Introducing the shifted time coordinate

$\bar{\tau} = \epsilon^m(\epsilon\tau - [\rho^{-2} - \rho^{-1}\phi])$ and rescaled variables $\bar{\beta} = \epsilon^n\beta$ and $\bar{\gamma} = \epsilon^p\gamma$ we find that the most structured balance occurs when $m = n = p = -1/2$.

The rescaled system is then,

$$\frac{d\bar{\beta}}{d\bar{\tau}} = -\bar{\beta}\bar{\gamma} + \rho(1 - 2\epsilon^{1/2}\bar{\beta})^2 \tag{4.21}$$

and

$$\frac{d\bar{\gamma}}{d\bar{\tau}} = -\rho\bar{\beta}\bar{\gamma}. \tag{4.22}$$

Expanding as $\bar{\beta} = \bar{\beta}_0 + \epsilon^{1/2}\bar{\beta}_1 + \cdots$ and similarly for $\bar{\gamma}$, the leading order problem is,

$$\frac{d\bar{\beta}_0}{d\bar{\tau}} = -\bar{\beta}_0\bar{\gamma}_0 + \rho \tag{4.23}$$

and

$$\frac{d\bar{\gamma}_0}{d\bar{\tau}} = -\rho\bar{\beta}_0\bar{\gamma}_0. \tag{4.24}$$

The system (4.24) can be rearranged to yield the Riccati equation [8],

$$\frac{d\bar{\gamma}_0}{d\bar{\tau}} = -(\rho^2\bar{\tau} + c_3 + \bar{\gamma}_0)\bar{\gamma}_0, \tag{4.25}$$

where $c_3$ is a constant. Seeking a solution of the form $\bar{\gamma}_0 = u'/u$ yields the separable equation $u'' = -(\rho^2\bar{\tau} + c_3)u'$, and hence,

$$\bar{\gamma}_0 = \frac{\exp(-\bar{\tau}(\rho^2\bar{\tau}/2 + c_3))}{\int_0^{\bar{\tau}} \exp(-s(\rho^2 s/2 + c_3))ds + \tilde{c}_4}. \tag{4.26}$$

The solution to the system can then be expressed in terms of the error function as

$$\bar{\beta}_0 = \frac{\exp(-\bar{\tau}(\rho^2\bar{\tau}/2 + c_3))}{\sqrt{\pi/2}\exp(c_3^2/2\rho^2)\left[\mathrm{erf}((\rho^2\bar{\tau} + c_3)/\rho\sqrt{2}) + c_4\right]} + \rho\bar{\tau} + \frac{c_3}{\rho} \tag{4.27}$$

and

$$\bar{\gamma}_0 = \frac{\rho\exp(-\bar{\tau}(\rho^2\bar{\tau}/2 + c_3))}{\sqrt{\pi/2}\exp(c_3^2/2\rho^2)\left[\mathrm{erf}((\rho^2\bar{\tau} + c_3)/\rho\sqrt{2}) + c_4\right]}. \tag{4.28}$$

The unknown constant $c_4$ can be fixed by considering the asymptotic form of $\bar{\gamma}_0$ as $\bar{\tau} \to -\infty$. Using the asymptotic form $\mathrm{erf}(x) \sim 1 - e^{-x^2}x^{-1}\pi^{-1/2}(1 + O(x^{-2}))$ and for brevity defining $f(\bar{\tau}) := (\rho^2\bar{\tau} + c_3)/\rho\sqrt{2}$, we have that

$$\mathrm{erf}(f(\bar{\tau})) = -1 - \frac{\exp(-f(\bar{\tau})^2)}{\sqrt{\pi}f(\bar{\tau})}(1 + O(f(\bar{\tau})^{-2})) \quad \text{as } \bar{\tau} \to -\infty. \tag{4.29}$$

Hence,

$$\bar{\gamma}_0(\bar{\tau}) = \frac{\rho\exp(-f(\bar{\tau})^2)}{\sqrt{\pi/2}\left[-1 - (\exp(-f(\bar{\tau})^2)/\sqrt{\pi}f(\bar{\tau}))(1 + O(f(\bar{\tau})^{-2})) + c_4\right]} \quad \text{as } \bar{\tau} \to -\infty. \tag{4.30}$$

Since $f(\bar{\tau}) \to -\infty$ as $\bar{\tau} \to -\infty$ we deduce that for $\bar{\gamma}_0(\bar{\tau})$ to tend to a non-zero limit we must have that $c_4 = 1$. In this case, the asymptotic behaviour is then

$$\bar{\gamma}_0(\bar{\tau}) = -\rho\sqrt{2}f(\bar{\tau})(1 + O(f(\bar{\tau})^{-2})) = -(\rho^2\bar{\tau} + c_3)(1 + O(\bar{\tau}^{-2})) \quad \text{as } \bar{\tau} \to -\infty. \tag{4.31}$$

In the region II variables, we then have

$$\gamma(T) = -\rho^2 T + (1 - \rho\phi) - \epsilon^{1/2}c_3 + O(\epsilon). \tag{4.32}$$

Matching to the region II solution then yields $c_3 = 0$.

The approximate solutions in region III in the original variables are therefore,

$$(\rho\bar{\tau})^2 = \epsilon^{-1}(\rho\epsilon\tau - [\rho^{-1} - \phi])^2,$$

$$\beta(\tau) = \epsilon^{1/2}\frac{\exp(-\epsilon^{-1}(\rho\epsilon\tau - [\rho^{-1} - \phi])^2/2)}{\sqrt{\pi/2}\left[\mathrm{erf}(\epsilon^{-1/2}(\rho\epsilon\tau - [\rho^{-1} - \phi])/\sqrt{2}) + 1\right]} + \rho\epsilon\tau - [\rho^{-1} - \phi] + O(\epsilon) \tag{4.33}$$

and

$$\gamma(\tau) = \epsilon^{1/2} \frac{\rho \exp\left(-\epsilon^{-1}(\rho\epsilon\tau - [\rho^{-1} - \phi])^2\right)}{\sqrt{\pi/2}\left[\mathrm{erf}\left(\epsilon^{-1/2}(\rho\epsilon\tau - [\rho^{-1} - \phi])/\sqrt{2}\right) + 1\right]} + O(\epsilon). \tag{4.34}$$

## 4.5. Summary of asymptotic solutions

In region I, $(\beta,\ \gamma,\ \tau = O(1))$,

$$\beta(\tau) = \frac{\phi(1 - \rho\phi)e^{(\rho\phi-1)\tau}}{1 - \rho\phi e^{(\rho\phi-1)\tau}} + O(\epsilon) \tag{4.35}$$

and

$$\gamma(\tau) = \frac{\rho\phi(1 - \rho\phi)e^{(\rho\phi-1)\tau}}{1 - \rho\phi e^{(\rho\phi-1)\tau}} + 1 - \rho\phi + O(\epsilon). \tag{4.36}$$

In region II, $(\beta = O(\epsilon),\ \gamma = O(1),\ \tau = O(\epsilon^{-1}))$,

$$\beta(\tau) = \frac{\epsilon\rho}{1 - \rho\phi - \rho^2\epsilon\tau} + O(\epsilon^2) \tag{4.37}$$

and

$$\gamma(\tau) = 1 - \rho\phi - \rho^2\epsilon\tau + O(\epsilon). \tag{4.38}$$

In region III, $(\beta = O(\epsilon^{1/2}),\ \gamma = O(\epsilon^{1/2}),\ \tau - \epsilon^{-1}[\rho^{-2} - \rho^{-1}\phi] = O(\epsilon^{-1/2}))$,

$$\beta(\tau) = \epsilon^{1/2} \frac{\exp\left(-\epsilon^{-1}(\rho\epsilon\tau - [\rho^{-1} - \phi])^2/2\right)}{\sqrt{\pi/2}\left[\mathrm{erf}\left(\epsilon^{-1/2}(\rho\epsilon\tau - [\rho^{-1} - \phi])/\sqrt{2}\right) + 1\right]} + \rho\epsilon\tau - [\rho^{-1} - \phi] + O(\epsilon) \tag{4.39}$$

and

$$\gamma(\tau) = \epsilon^{1/2} \frac{\rho \exp\left(-\epsilon^{-1}(\rho\epsilon\tau - [\rho^{-1} - \phi])^2\right)}{\sqrt{\pi/2}\left[\mathrm{erf}\left(\epsilon^{-1/2}(\rho\epsilon\tau - [\rho^{-1} - \phi])/\sqrt{2}\right) + 1\right]} + O(\epsilon). \tag{4.40}$$

In region IV, $(\beta = O(1),\ \gamma = O(\epsilon^n)$ for all $n > 0,\ \tau = O(\epsilon^{-1}))$,

$$\beta(\tau) = \frac{1}{2} - \frac{1}{2(1 + 2[\phi - \rho^{-1} + \rho\epsilon\tau])} \tag{4.41}$$

and

$$\gamma(\tau) = O(\epsilon^n), \quad \text{all } n > 0, \tag{4.42}$$

the error being beyond any algebraic term in $\epsilon$. A combined plot of all of the four asymptotic solutions is given in figure 3, showing $(a,b)$ the time course of each of $\beta$ and $\gamma$ against a logarithmic time axis, and also $(c)$ in the $(\beta,\ \gamma)$–phase plane, for comparison with figures 1 and 2.

## 4.6. Comparison of asymptotic and numerical approximations

A comparison of the asymptotic approximations against a numerical solution of the system (calculated with the stiff solver `ode15s`, Matlab, Mathworks) is shown in figure 4, with the reaction rate ratio $\epsilon$ taken as 0.001, reactants ratio $\rho = 2$ and iodide : iodine ratio $\phi = 0.2$ (these values are chosen arbitrarily). The region I solution follows the numerical approximation very closely up to around $\tau = 5$. The region II solution then follows the numerical solution to within a few per cent up to around $\tau = 100$. The region III solution then follows the numerical solution closely around the dimensionless switchover time $\epsilon^{-1}[\rho^{-2} - \rho^{-1}\phi] = 150$, up to around $\tau = 200$. Finally, the agreement between the region IV solution for $\tau > 200$ is excellent, as would be expected from the smaller-than-algebraic error in equations (4.41) and (4.42).

# 5. Experiments

In this section, we present the results of some 'kitchen sink' experiments, conducted with readily available chemicals. The protocol is essentially as described by Wright [14] (Procedure D. Reaction Using Kitchen Measuring Ware) only with 3% Lugol's iodine instead of tincture of iodine 2%, and varying the quantities of water and iodine used.

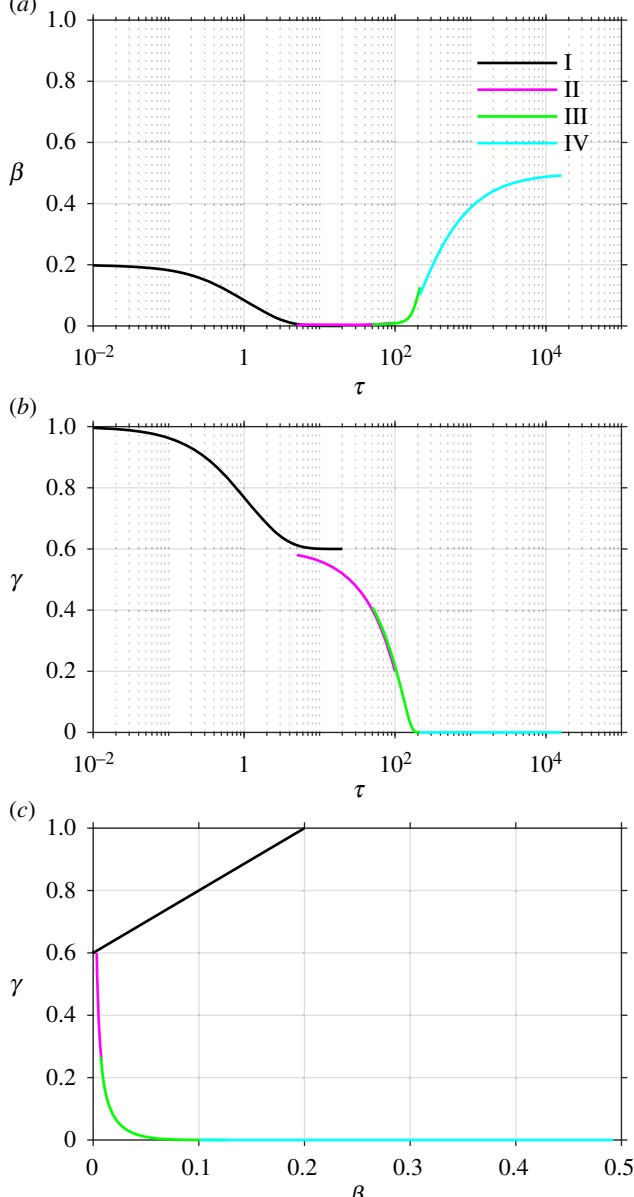

**Figure 3.** Asymptotic solutions plotted ($a$,$b$) versus dimensionless time $\tau$ (logarithmic scale), ($c$) in the $\beta$, $\gamma$-phase plane, colour-coded to match figure 1. Dimensionless groups are chosen as $\rho = 2$, $\epsilon = 0.001$, $\phi = 0.2$.

Vitamin C stock solution was prepared by mixing a 1000 mg vitamin C tablet with 30–120 ml water. Solution 'A' is prepared from 5 ml (1 teaspoon) of vitamin C stock solution, between 2.5 and 10 ml of 3% w/v Lugol's iodine, and 60 ml (4 tablespoons) of warm tap water at 40°C. Solution 'B' is prepared from 15 ml of 3% hydrogen peroxide, 1 level teaspoon of powdered laundry starch and 60 ml of warm water. Solutions A and B are added in a beaker, at which point a timer is started, and the mixture is stirred manually for 5 s. The timer is stopped when the colour change from clear to blue occurs.

Lugol's iodine is formulated as a 1 : 2 mixture of iodine ($I_2$) and potassium iodide (KI) [20]. Based on molar masses of 126.9 g mol$^{-1}$ for iodine and 166 g mol$^{-1}$ for potassium iodide, Lugol's 3% w/v solution contains $1 + 2 \times 126.9/166 = 2.5289$ g/100 ml of iodine. The range 2.5–10 ml of 3% w/v Lugol's therefore corresponds to between 0.0632 and 0.2529 g of iodine, i.e. 0.4982–1.9928 mmol. The initial concentration of molecular iodine in Lugol's is unknown, so the quantity $\phi = b_0/m_0$ will be determined alongside the reaction rate via parameter fitting.

Vitamin C has a molar mass of 176.12 g mol$^{-1}$ and hence a 1000 mg tablet diluted in 30–120 ml water yields stock solutions with a concentration of 0.18926–0.04731 mol l$^{-1}$. Hence 5 ml contains 0.9463–0.2366

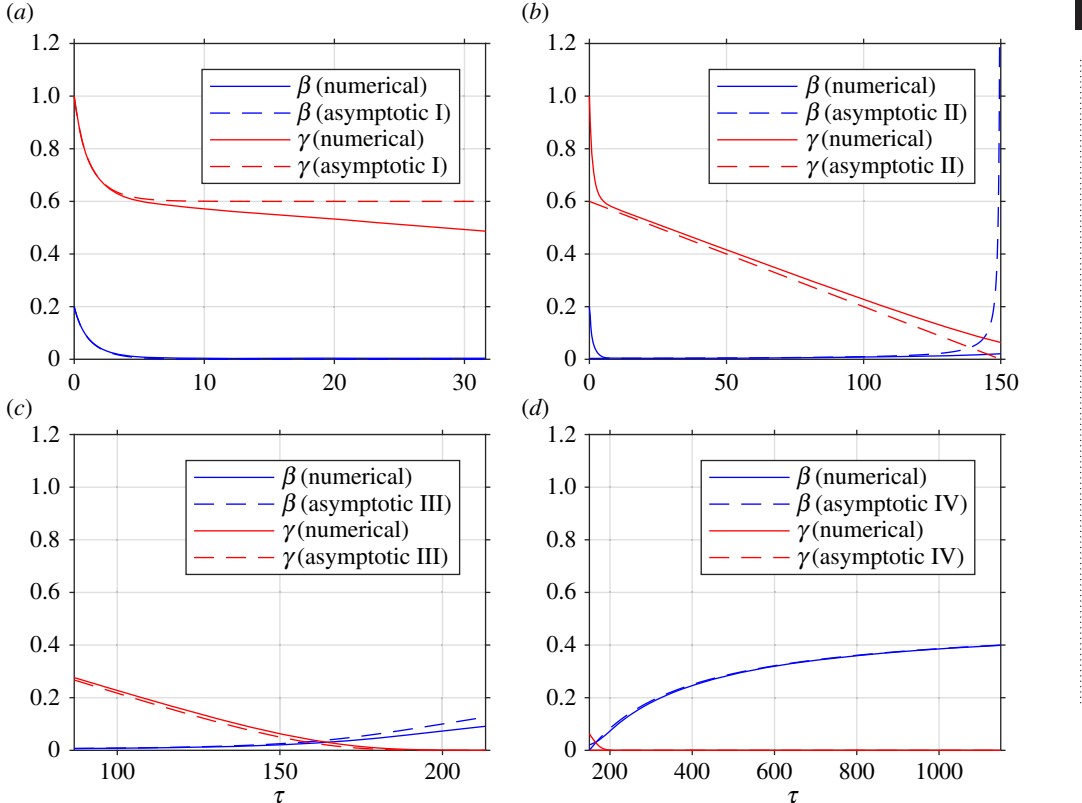

**Figure 4.** Comparison between asymptotic and numerical solutions with $\epsilon = 0.001$, $\rho = 2$, $\phi = 0.2$, corresponding to dimensionless switchover time $\tau = 150$. (a) Region I, (b) region II, (c) region III, (d) region IV.

**Table 1.** Switchover time experimental results for (a) variable vitamin C concentration, (b) variable iodine concentration. Results show two repeats.

| Vit. C dil. (ml) | Vit. C stock conc. (mol l$^{-1}$) | Lugol's (ml) | $c_0$ (mol l$^{-1}$) | $m_0$ (mol l$^{-1}$) | $t_{sw}$ (s) |
|---|---|---|---|---|---|
| (a) | | | | | |
| 120 | 0.04731 | 5 | 0.001632 | 0.0068718 | (48.12, 43.56) |
| 90 | 0.06309 | 5 | 0.002175 | 0.0068718 | (68.72, 74.32) |
| 60 | 0.09463 | 5 | 0.003263 | 0.0068718 | (116.86, 122.34) |
| 45 | 0.12618 | 5 | 0.004351 | 0.0068718 | (153.03, 168.08) |
| 30 | 0.18926 | 5 | 0.006526 | 0.0068718 | (243.67, 210.94) |
| (b) | | | | | |
| 60 | 0.09463 | 2.5 | 0.003320 | 0.0034962 | (426.97, 495.78) |
| 60 | 0.09463 | 3.75 | 0.003292 | 0.0051987 | (285.42, 246.28) |
| 60 | 0.09463 | 5 | 0.003263 | 0.0068718 | (116.86, 122.34) |
| 60 | 0.09463 | 7.5 | 0.003208 | 0.0101330 | (55.71, 43.71) |
| 60 | 0.09463 | 10 | 0.003154 | 0.0132855 | (19.51, 22.64) |

mmol of vitamin C. Concentrations are calculated based on 120 ml water, 5 ml stock, 15 ml peroxide and 2.5–10 ml Lugol's. The concentration of hydrogen peroxide in all experiments is 0.098 mol l$^{-1}$.

Results for two series, (a) varying vitamin C with iodine held fixed, and (b) varying iodine with vitamin C held fixed, are shown in table 1. The outcome of unconstrained least-squares fitting equation (4.13) for the parameters $k_0$ and $\phi$ to both experimental series simultaneously is shown in figure 5; we find $k_0 \approx 0.57\,\mathrm{M}^{-1}\,\mathrm{s}^{-1}$ and $\phi = -7 \times 10^{-5} \approx 0$.

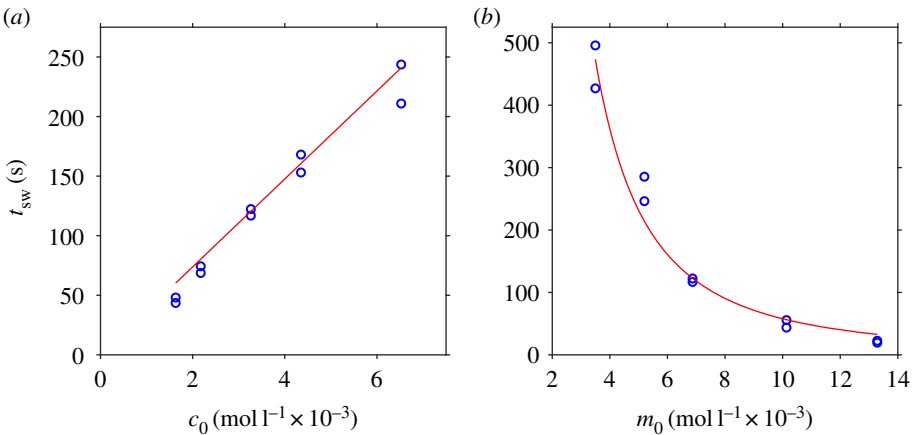

**Figure 5.** Experimental results (blue circles) from table 1 and parameter fit to equation (4.13) with $k_0 = 0.57$ M$^{-1}$ s$^{-1}$ and $\phi \approx 0$ (red lines). (a) Varying $c_0$ with $m_0 = 0.0068718$ mol l$^{-1}$. (b) Varying $m_0$ with $c_0 \approx 0.033$ mol l$^{-1}$.

## 6. Discussion

Clock reactions have long provided an instructive pedagogical example in chemistry education, in addition to their industrial and biochemical importance. This study focused on the vitamin C clock reaction, the dynamics of which are governed by substrate-depletion without autocatalysis. A fast vitamin C-dependent reaction converts iodine to iodide, depleting vitamin C in the process. A slow vitamin C-independent reaction converts iodide back to iodine. During a long induction period, the fast reaction dominates and very little molecular iodine is present in comparison to iodide. Once the vitamin C supply is exhausted, the slow reaction takes over and the molecular iodine level rises, which can be visualized by a colour change in the presence of starch.

This system can be described effectively via techniques of matched asymptotic expansions, previously used successfully for the indirectly catalytic iodate−arsenous acid reaction [8] in addition to other systems. This analysis enables the construction of an approximate solution, and moreover a compact expression for the switchover time which depends on the initial concentrations of vitamin C and iodine, and the rate of the slow reaction, i.e.

$$t_{sw} = \frac{[C_6H_8O_6]_0 - [I_2]_0}{(2[I_2]_0 + [I^-]_0)^2 k_0}. \tag{6.1}$$

Results from 'kitchen sink chemistry' experiments were found to follow this model very closely, with the parameters $k_0$ and $\phi$ being fitted simultaneously to data series varying in each of $c_0$ and $m_0$. The fitted parameter representing initial molecular iodine proportion $\phi$ was approximately zero.

The analysis presented here should be of interest in both applications involving substrate-depletion dynamics, and also for pedagogical purposes in mathematical chemistry. The core ideas employed: law of mass action, dimensional analysis, quasi-steady kinetics, phase planes, matched asymptotics, numerical solutions and parameter fitting, are central to mathematical biology and chemistry, and within the reach of advanced undergraduates and masters students in mathematical modelling. Moreover, the experiment can be carried out using relatively safe chemicals and minimal equipment. A potential interesting avenue for further investigation, also accessible to student modellers, would be to vary the temperature of the reactants and assess whether the change to switchover time may be predicted via an Arrhenius equation for $k_0$. A further topic to explore would be to use techniques from analytical chemistry such as UV−visible absorbance [19] to measure the trajectory of the solution quantitatively and compare to the mathematical solution. Clock reactions provide enduring and accessible examples of mathematical modelling and experiment.

Data accessibility. Data are available at the Dryad Digital Repository: https://datadryad.org/resource/doi:10.5061/dryad.68q4hf7 [21].

Competing interests. We declare we have no competing interests.

Authors' contributions. W.M.T. and R.K. carried out the mathematical modelling and analysis with support from D.J.S. R.K. conducted the experiments and parameter fitting. D.J.S. supervised the research and wrote the paper.

Funding. D.J.S. acknowledges Engineering and Physical Sciences Research Council award no. EP/N021096/1.

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
