## [Reviewer comments · Royal Society Open Science]

Review History

RSOS-181367.R0 (Original submission)

Review form: Reviewer 1

Is the manuscript scientifically sound in its present form?

No

Are the interpretations and conclusions justified by the results?

No

Is the language acceptable?

Yes

Is it clear how to access all supporting data?

Yes

Do you have any ethical concerns with this paper?

No

Have you any concerns about statistical analyses in this paper?

No

Recommendation?

Reject

Comments to the Author(s)

See attached file (Appendix A).

Review form: Reviewer 2

Is the manuscript scientifically sound in its present form?

Yes

Are the interpretations and conclusions justified by the results?

Yes

Is the language acceptable?

Yes

Is it clear how to access all supporting data?

Yes

Do you have any ethical concerns with this paper?

No

Have you any concerns about statistical analyses in this paper?

No

Recommendation?

Accept with minor revision (please list in comments)

Comments to the Author(s)

I found this manuscript easy to read and the conclusion is well-supported by the data and analysis.

Minor issues needed to be corrected are listed as follows:

1. The author states in the abstract that they found an approximate formula between the switchover time and the concentration of the reactants and the rate coefficient of the slow reaction. The analysis has however been performed in dimensionless form. It would also be proper to see this formula directly like in case of a Landolt reaction (see: A. K. Horvath et al, J. Phys. Chem. A, 2008, 112, 7868.) This would be helpful and useful for a general reader who is rather a chemist than a mathematician.
2. Page 2, line 6 from bottom: the term "iodine-thiosulfate system" used is certainly incorrect because this is a very rapid reaction. This system the authors would have wanted to refer is the iodide-peroxydisulfate system in presence of thiosulfate.
3. Page 11, last sentence prior to discussion: k_{AB} is a rate constant of a second order reaction as shown in eq. (1.2) therefore its unit has to be $M^{-1}s^{-1}$ instead of s^{-1} . It definitely has to be corrected.
4. It has to be mentioned somewhere in the paper that hydrogen-peroxide is used in a huge

excess compared to iodine an iodide. This will allow to use eq. (1.2) as is, otherwise the concentration of hydrogen-peroxide has to be included in the rate equation.

5. It is also important to know the exact concentration of H₂O₂ in order to check the validity of data in Table 1a and 1b. This info in terms of molarity has to be given somewhere in the article.

My final evaluation is therefore that this manuscript requires a minor revisions prior to acceptance.

Review form: Reviewer 3

Is the manuscript scientifically sound in its present form?

Yes

Are the interpretations and conclusions justified by the results?

Yes

Is the language acceptable?

Yes

Is it clear how to access all supporting data?

Yes

Do you have any ethical concerns with this paper?

No

Have you any concerns about statistical analyses in this paper?

No

Recommendation?

Accept with minor revision (please list in comments)

Comments to the Author(s)

Clock reactions are of general interest in teaching kinetics and in some research applications. Here a mathematical analysis is undertaken of the basic mechanism of an inhibitor clock reaction resulting in a simple expression for clock times, or switchover times, that are compared to experimental results with the Vitamin C clock reaction.

I have some points for consideration before publication:

It would be clearer if the rate constants subscripts were the reaction number eg k_2 or $k_{1.2}$ rather than k_{AB} and k_{BA} .

It is not mentioned, but in 1.2 why are you are assuming acid is constant, or that rate is independent of acid? What is the source of H⁺ once all the vitamin C is consumed?

Figure 1. (b) and (c) are indicated as showing close-ups, but the scale is the same on the x axis, it looks like more vectors have been added. Why are they smaller in magnitude at the same position in (a)?

Pg10 (f) mentions $\phi = \text{iodate}:\text{iodine ratio}$ but this is the iodine I₂ to total iodine atoms (I⁻ + I₂). The discussion also includes conversion of iodine to iodate. I think iodide (I⁻) is meant instead of iodate (IO₃⁻).

Pg 11. Misleading to say all initial iodine molecules are dissociated. In lugol's reagent iodine mainly reacts with iodide to form triiodide I₃⁻ (source of I₂). However, you prepare a solution A with vitamin C and lugol's so you are assuming the reaction with vitamin C and the iodine in

Iugol's has occurred producing hydrogen ion and iodide?

Perhaps best to write iodine atoms for total iodine to distinguish from iodine molecules – then it is 2.5289 g /100 ml of iodine atoms in the experimental section (not iodine (I₂)).

Why are concentrations are calculated based on 5 ml starch? It is added as 1 teaspoon powder so shouldn't affect the volume.

Decision letter (RSOS-181367.R0)

04-Jan-2019

Dear Professor Smith,

The editors assigned to your paper ("Mathematical modelling of the vitamin C clock reaction") have now received comments from reviewers. We would like you to revise your paper in accordance with the referee and Associate Editor suggestions which can be found below (not including confidential reports to the Editor). Please note this decision does not guarantee eventual acceptance.

Please submit a copy of your revised paper before 27-Jan-2019. Please note that the revision deadline will expire at 00.00am on this date. If we do not hear from you within this time then it will be assumed that the paper has been withdrawn. In exceptional circumstances, extensions may be possible if agreed with the Editorial Office in advance. We do not allow multiple rounds of revision so we urge you to make every effort to fully address all of the comments at this stage. If deemed necessary by the Editors, your manuscript will be sent back to one or more of the original reviewers for assessment. If the original reviewers are not available, we may invite new reviewers.

- Data accessibility

It is a condition of publication that all supporting data are made available either as supplementary information or preferably in a suitable permanent repository. The data

accessibility section should state where the article's supporting data can be accessed. This section should also include details, where possible of where to access other relevant research materials such as statistical tools, protocols, software etc can be accessed. If the data have been deposited in an external repository this section should list the database, accession number and link to the DOI for all data from the article that have been made publicly available. Data sets that have been deposited in an external repository and have a DOI should also be appropriately cited in the manuscript and included in the reference list.

If you wish to submit your supporting data or code to Dryad (<http://datadryad.org/>), or modify your current submission to dryad, please use the following link:
<http://datadryad.org/submit?journalID=RSOS&manu=RSOS-181367>

- **Competing interests**

- **Authors' contributions**

- **Acknowledgements**

- **Funding statement**

on behalf of Mr Andrew Dunn (Associate Editor) and Mark Chaplain (Subject Editor)
openscience@royalsociety.org

Associate Editor's comments:

As there is a split in opinion between the reviewers here, the Editors have opted to give the authors the benefit of the doubt. However, you must ensure that your revision fully addresses the concerns of the reviewers (in particular reviewer 1) before you resubmit. If you do not incorporate responses to their concerns or at least provide a full rebuttal to them, we will not be able to consider further revisions of the manuscript.

Comments to Author:

Reviewers' Comments to Author:

Reviewer: 1

Comments to the Author(s)

See attached file

Reviewer: 2

Comments to the Author(s)

I found this manuscript easy to read and the conclusion is well-supported by the data and analysis.

Minor issues needed to be corrected are listed as follows:

1. The author states in the abstract that they found an approximate formula between the switchover time and the concentration of the reactants and the rate coefficient of the slow reaction. The analysis has however been performed in dimensionless form. It would also be proper to see this formula directly like in case of a Landolt reaction (see: A. K. Horvath et al, J. Phys. Chem. A, 2008, 112, 7868.) This would be helpful and useful for a general reader who is rather a chemist than a mathematician.
2. Page 2, line 6 from bottom: the term "iodine-thiosulfate system" used is certainly incorrect because this is a very rapid reaction. This system the authors would have wanted to refer is the iodide-peroxydisulfate system in presence of thiosulfate.
3. Page 11, last sentence prior to discussion: k_{AB} is a rate constant of a second order reaction as shown in eq. (1.2) therefore its unit has to be $M^{-1}s^{-1}$ instead of s^{-1} . It definitely has to be corrected.
4. It has to be mentioned somewhere in the paper that hydrogen-peroxide is used in a huge excess compared to iodine an iodide. This will allow to use eq. (1.2) as is, otherwise the concentration of hydrogen-peroxide has to be included in the rate equation.
5. It is also important to know the exact concentration of H₂O₂ in order to check the validity of data in Table 1a and 1b. This info in terms of molarity has to be given somewhere in the article.

My final evaluation is therefore that this manuscript requires a minor revisions prior to acceptance.

Reviewer: 3

Comments to the Author(s)

Clock reactions are of general interest in teaching kinetics and in some research applications. Here a mathematical analysis is undertaken of the basic mechanism of an inhibitor clock reaction resulting in a simple expression for clock times, or switchover times, that are compared to experimental results with the Vitamin C clock reaction.

I have some points for consideration before publication:

It would be clearer if the rate constants subscripts were the reaction number eg k_2 or $k_{1.2}$ rather than k_{AB} and k_{BA} .

It is not mentioned, but in 1.2 why are you assuming acid is constant, or that rate is independent of acid? What is the source of H^+ once all the vitamin C is consumed?

Figure 1. (b) and (c) are indicated as showing close-ups, but the scale is the same on the x axis, it looks like more vectors have been added. Why are they smaller in magnitude at the same position in (a)?

Pg10 (f) mentions ϕ = iodate:iodine ratio but this is the iodine I_2 to total iodine atoms ($I^- + I_2$). The discussion also includes conversion of iodine to iodate. I think iodide (I^-) is meant instead of iodate (IO_3^-).

Pg 11. Misleading to say all initial iodine molecules are dissociated. In lugol's reagent iodine mainly reacts with iodide to form triiodide I_3^- (source of I_2). However, you prepare a solution A with vitamin C and lugol's so you are assuming the reaction with vitamin C and the iodine in lugol's has occurred producing hydrogen ion and iodide?

Perhaps best to write iodine atoms for total iodine to distinguish from iodine molecules - then it is 2.5289 g /100 ml of iodine atoms in the experimental section (not iodine (I_2)).

Why are concentrations are calculated based on 5 ml starch? It is added as 1 teaspoon powder so shouldn't affect the volume.

Author's Response to Decision Letter for (RSOS-181367.R0)

See Appendices B & C.

RSOS-181367.R1 (Revision)

Review form: Reviewer 1

Is the manuscript scientifically sound in its present form?

No

Are the interpretations and conclusions justified by the results?

No

Is the language acceptable?

Yes

Is it clear how to access all supporting data?

Yes

Do you have any ethical concerns with this paper?

No

Have you any concerns about statistical analyses in this paper?

No

Recommendation?

Reject

Comments to the Author(s)

See attached file (Appendix D).

Review form: Reviewer 2**Is the manuscript scientifically sound in its present form?**

Yes

Are the interpretations and conclusions justified by the results?

Yes

Is the language acceptable?

Yes

Is it clear how to access all supporting data?

Not Applicable

Do you have any ethical concerns with this paper?

No

Have you any concerns about statistical analyses in this paper?

No

Recommendation?

Accept as is

Comments to the Author(s)

The authors have successfully addressed my concerns therefore I recommend this paper for publication.

Review form: Reviewer 4**Is the manuscript scientifically sound in its present form?**

Yes

Are the interpretations and conclusions justified by the results?

Yes

Is the language acceptable?

Yes

Is it clear how to access all supporting data?

Yes

Do you have any ethical concerns with this paper?

No

Have you any concerns about statistical analyses in this paper?

No

Recommendation?

Accept as is

Comments to the Author(s)

I completely agree with the authors, Reviewer 1 has made no scientific case for rejecting the paper. The arguments by the authors, in particular in reply to Reviewer 1's concerns about the reaction rate modelling, are rational, reasonable and supported by the experimental evidence. I believe that the paper should be published.

Decision letter (RSOS-181367.R1)

07-Mar-2019

Dear Professor Smith,

I am pleased to inform you that your manuscript entitled "Mathematical modelling of the vitamin C clock reaction" is now accepted for publication in Royal Society Open Science.

Kind regards,

Andrew Dunn

on behalf of Prof Mark Chaplain (Subject Editor)

Associate Editor Comments to Author:

Following your appeal, the Editors invited a third referee to assess the manuscript, and your response to the original rejection decision. The referee has found in your favour, and the Editors agree that your manuscript may now be published 'as is'. Thank you for your support of the journal.

Reviewer comments to Author:

Reviewer: 1

Comments to the Author(s)

See attached file

Reviewer: 2

Comments to the Author(s)

The authors have successfully addressed my concerns therefore I recommend this paper for publication.

Reviewer: 4

Comments to the Author(s)

I completely agree with the authors, Reviewer 1 has made no scientific case for rejecting the paper. The arguments by the authors, in particular in reply to Reviewer 1's concerns about the reaction rate modelling, are rational, reasonable and supported by the experimental evidence. I believe that the paper should be published.

Appendix A

The manuscript 'Mathematical modelling of the vitamin C clock reaction' by Kerr, Thomson, and Smith, RSOS-181367, is a very interesting study on the mathematical modeling of chemical reactions, specially for clock reactions. The study is deep and explore several aspects of the mathematical modeling of chemical reactions. The mathematical concepts looks correct. However, the manuscript has a problem with the rate law of the reaction between hydrogen peroxide and iodide, equation (1.2). The chemical reaction

has a well established rate law*

$$\text{rate} = k[\text{I}^-][\text{H}_2\text{O}_2] \quad (\text{in buffered media})$$

$$\text{rate} = k[\text{I}^-][\text{H}_2\text{O}_2] + k'[\text{I}^-][\text{H}_2\text{O}_2][\text{H}^+] \quad (\text{considering the acid catalysis})$$

In this way, the rate law indicated by eq (1.2)

$$\text{rate} = k[\text{I}^-]^2$$

cannot be accepted, which turns the mathematical work on this reaction unrealistic.

As an additional comment, I suggest that, at discretion of the authors, in the Introduction section, the work by G. Lente, G. Bazsa, and I. Fabian (What is and what isn't a clock reaction? *New J. Chem.* 31:1707, 2007) should be cited and discussed because the definition of a clock reaction is not settled completely yet.

*References for the rate law of reaction (1.1)

1. H. A. Liebhasfky and A. Mohammad. The Kinetics of the Reduction, in Acid Solution, of Hydrogen Peroxide by Iodide Ion. *J. Am. Chem. Soc.*, 55(10):3977-3986, 1933.
2. C. L Copper and E. Koubek. A Kinetics Experiment To Demonstrate the Role of a Catalyst in a Chemical Reaction: A Versatile Exercise for General or Physical Chemistry Students. *J. Chem. Educ.*, 75(1):87-89, 1998.
3. P. D. Sattsangi. A Microscale Approach to Chemical Kinetics in the General Chemistry Laboratory: The Potassium Iodide Hydrogen Peroxide Iodine-Clock Reaction. *J. Chem. Educ.*, 88(2):184-188, 2011.

Appendix B

Mathematical modelling of the vitamin C clock reaction

RSOS-181367

Response to reviewers' comments

Ryan Kerr, William Thomson and David Smith

January 26, 2019

We thank the reviewers for their careful reading of the manuscript and thought-provoking comments, and we thank the editors for the opportunity to submit a revised manuscript. The query of reviewer 1 regarding the correct rate law for the forward reaction was the most significant issue to address, and it prompted us to review the model and relevant literature carefully. We have responded below in detail to this query, which we believe we can address, and to all other issues raised (reviewers comments in italics, our responses in roman).

1 Reviewer 1

The manuscript Mathematical modelling of the vitamin C clock reaction by Kerr, Thomson, and Smith, RSOS-181367, is a very interesting study on the mathematical modeling of chemical reactions, specially for clock reactions. The study is deep and explore several aspects of the mathematical modeling of chemical reactions. The mathematical concepts looks correct.

However, the manuscript has a problem with the rate law of the reaction between hydrogen peroxide and iodide, equation (1.2). The chemical reaction

has a well established rate law^{1,2,3}

$$\text{rate} = k[I^-][H_2O_2] \quad (\text{in buffered media})$$

$$\text{rate} = k[I^-][H_2O_2] + kN[I^-][H_2O_2][H^+] \quad (\text{considering the acid catalysis})$$

In this way, the rate law indicated by eq (1.2)

$$\text{rate} = k[I^-]^2$$

cannot be accepted, which turns the mathematical work on this reaction unrealistic.

This is a significant and thought-provoking comment which, along with Reviewer 2’s queries regarding the hydrogen peroxide concentration, led us to re-evaluate the model.

In view of the cited references, we re-worked the calculation using the linear rate law indicated above. The asymptotics are fairly similar with a few details of the solutions changed. The key difference is that expression for the switchover time under linear kinetics is of the form,

$$t_{sw} \propto m_0^{-1},$$

as compared with our submitted model $t_{sw} \propto m_0^{-2}$.

Taking a log-log plot of the experimental data series with m_0 varying results in a gradient of approximately -2.3 , which is much closer to the initial model, moreover while the initial model fits our experimental data well, the new model does not, as shown below (best one-parameter fit for k_0 across two data series simultaneously):

We believe the reason that the original model works better than the suggested linear law is that in our experiments there is a great excess of hydrogen peroxide in our experiment (0.09 mol/l as compared with 0.001–0.006 mol/l vitamin C – we have now included this detail in section 5). This excess ensures that the rate-limiting step (nucleophilic attack of I^- upon the H_2O_2 in equation (4) of Copper and Koubek, 1998) underlying the linear rate laws is no longer rate-limiting, and the reaction can instead be modelled by a quadratic law. By contrast, references^{1,2,3} worked with concentrations of H_2O_2 which are much closer to those of the other substrates.

We therefore believe that (without undertaking a much more complex model involving multiple reaction steps) the original rate law of $k_0[I^-]^2$ provides a suitable approximation which is validated by our experiments. We have however modified the introduction (after equation 1.1) to include the suggested references and to clarify our reasoning for taking the quadratic rate law, and moreover have reported the value of the hydrogen peroxide concentration in section 5 (4th paragraph).

As an additional comment, I suggest that, at discretion of the authors, in the Introduction section, the work by G. Lente, G. Bazsa, and I. Fabian (What is and what isn’t a clock reaction? New J. Chem. 31:1707, 2007) should be cited and discussed because the definition of a clock reaction is not settled completely yet.

Thank you for the suggestion – we now refer to this work at the end of the fourth paragraph of the introduction (following the comments about autocatalysis), and in the last paragraph of the introduction (highlighting that the reaction we consider fits the strict

definition of clock reaction given by Lente et al.)

References

1. H. A. Liebhasfky and A. Mohammad. The Kinetics of the Reduction, in Acid Solution, of Hydrogen Peroxide by Iodide Ion. *J. Am. Chem. Soc.*, 55(10):3977-3986, 1933.
2. C. L. Copper and E. Koubek. A Kinetics Experiment To Demonstrate the Role of a Catalyst in a Chemical Reaction: A Versatile Exercise for General or Physical Chemistry Students. *J. Chem. Educ.*, 75(1):87-89, 1998.
3. P. D. Sattsangi. A Microscale Approach to Chemical Kinetics in the General Chemistry Laboratory: The Potassium Iodide Hydrogen Peroxide Iodine-Clock Reaction. *J. Chem. Educ.*, 88(2):184-188, 2011.

2 Reviewer 2

I found this manuscript easy to read and the conclusion is well-supported by the data and analysis. Minor issues needed to be corrected are listed as follows:

1. *The author states in the abstract that they found an approximate formula between the switchover time and the concentration of the reactants and the rate coefficient of the slow reaction. The analysis has however been performed in dimensionless form. It would also be proper to see this formula directly like in case of a Landolt reaction (see: A. K. Horvath et al, *J. Phys. Chem. A*, 2008, 112, 7868.) This would be helpful and useful for a general reader who is rather a chemist than a mathematician.*

We have now reported the dimensional form of the switchover time result explicitly in terms of chemical concentrations in equation (6.1) in the discussion, in a similar form to Horváth et al.

2. *Page 2, line 6 from bottom: the term “iodine-thiosulfate system” used is certainly incorrect because this is a very rapid reaction. This system the authors would have wanted to refer is the iodide-peroxydisulfate system in presence of thiosulfate.*

Thank you for identifying this error, we have corrected (last paragraph of section 1) as suggested.

3. *Page 11, last sentence prior to discussion: k_{AB} is a rate constant of a second order reaction as shown in eq. (1.2) therefore its unit has to be $M^{-1}s^{-1}$ instead of s^{-1} . It definitely has to be corrected.*

Thank you – corrected.

4. *It has to be mentioned somewhere in the paper that hydrogen-peroxide is used in a huge excess compared to iodine an iodide. This will allow to use eq. (1.2) as is, otherwise the concentration of hydrogen-peroxide has to be included in the rate equation.*

Agreed – we have added comments about this issue before equation (1.2) (please see also the detailed response to Reviewer 1’s query about the rate law above and response to the next comment).

5. *It is also important to know the exact concentration of H₂O₂ in order to check the validity of data in Table 1a and 1b. This info in terms of molarity has to be given somewhere in the article.*

This concentration is now given in the last but one paragraph of section 5. The quantity is 0.09 mol/l, which is indeed an order of magnitude greater than c_0 and m_0 .

3 Reviewer 3

Clock reactions are of general interest in teaching kinetics and in some research applications. Here a mathematical analysis is undertaken of the basic mechanism of an inhibitor clock reaction resulting in a simple expression for clock times, or switchover times, that are compared to experimental results with the Vitamin C clock reaction. I have some points for consideration before publication:

- *It would be clearer if the rate constants subscripts were the reaction number e.g. k_2 or $k_{1,2}$ rather than k_{AB} and k_{BA} .*

We have considered this suggestion and have taken on board as follows: a numbered scheme as been used as suggested, with the nomenclature k_0 for the forward reaction (formerly k_{AB}) and k_1 for the backward reaction (formerly k_{BA}), consistent with Billingham and Needham, Phil. Trans. R. Soc. (1992) 340, 569–591, section 3.

- *It is not mentioned, but in 1.2 why are you are assuming acid is constant, or that rate is independent of acid? What is the source of H⁺ once all the vitamin C is consumed?*

We have assumed that the production of H^+ by reaction 1.2 at the same rate as I^- ensures that there is then sufficient H^+ available for reaction 1.1 to proceed once the vitamin C is consumed. A brief note has been added before equation 1.3 to this effect.

- *Figure 1. (b) and (c) are indicated as showing close-ups, but the scale is the same on the x axis, it looks like more vectors have been added. Why are they smaller in magnitude at the same position in (a)?*

Good point! Figure 1 (b) and (c) additionally use a rescaling for the arrows, which results in arrows which are very small in (a) appearing visible – we have added a note in the caption clarifying that the arrows have been rescaled.

- *Pg10 (f) mentions phi = iodate:iodine ratio but this is the iodine I₂ to total iodine atoms (I⁻ + I₂). The discussion also includes conversion of iodine to iodate. I think iodide (I⁻) is meant instead of iodate (IO₃⁻).*

Agreed – we have corrected this in both page 10 (f) and also the discussion.

- *Pg 11. Misleading to say all initial iodine molecules are dissociated. In lugols reagent iodine mainly reacts with iodide to form triiodide I3- (source of I2). However, you prepare a solution A with vitamin C and lugols so you are assuming the reaction with vitamin C and the iodine in lugols has occurred producing hydrogen ion and iodide?*

This comment led us to revise the data analysis. We now agree that we cannot *a priori* assume that the iodine is dissociated initially. We do however wish to base the value of ϕ on the molecular iodine proportion before the reaction with vitamin C starts, and given that the equilibrium dynamics of iodine/iodide in Lugol's is beyond the scope of the paper, we decided instead to take ϕ as a free parameter. Hence the fitting process now finds both k_0 and ϕ .

Interestingly, the optimal value of ϕ from an unconstrained optimization is smaller than 10^{-4} , suggesting that the initial proportion of molecular iodine is very small.

We have hence modified the abstract (no longer referring to a 'single-parameter fit'), have modified the statements regarding the value of ϕ in the third paragraph of section 5, added the fitted value of ϕ to the last paragraph of section 5 and caption of figure 5, and modified the second paragraph of the discussion.

- *Perhaps best to write iodine atoms for total iodine to distinguish from iodine molecules then it is 2.5289 g /100 ml of iodine atoms in the experimental section (not iodine (I2)).*

In this case (measuring g/ml rather than mol/l) we believe the distinction between atoms and molecules isn't necessary, as two atoms of iodine will have (virtually) the same mass as one molecule of I_2 – therefore we haven't made a change.

- *Why are concentrations are calculated based on 5 ml starch? It is added as 1 teaspoon powder so shouldn't affect the volume.*

We agree with the reviewer and have removed the 5 ml of liquid starch from the calculation. This slightly alters the concentration values and hence table 1, figure 5 and the fitted value of k_0 have been updated (the latter changes from 0.59 to 0.57).

4 Other changes

On revising the manuscript we have made a small number of minor corrections/clarifications to the mathematical arguments, listed below:

- Typographical error: equation (4.7) was missing an ϵ in front of the left hand side term. This has been corrected.
- We have corrected and clarified the argument around equations (4.16)–(4.17).
- Rewritten the mathematical expression for $\text{erf}(x)$ just after equation (4.28) for clarity, and correcting (improving) the error term from $O(x^{-1})$ to $O(x^{-2})$.
- Explicitly including the error term in equations (4.29)–(4.31), and corrected (improved) the error term in (4.32) to $O(\epsilon)$.

Appendix C

Mathematical modelling of the vitamin C clock reaction RSOS-181367 Rebuttal to Reviewer 1

Ryan Kerr, William Thomson and David Smith

February 20, 2019

We thank the Editors and appointed Adjudicator for the opportunity to appeal Reviewer 1's decision.

Background

The point of contention is the correct rate law for the reaction

which produces iodine from iodide.

Our first submission applied the law of mass action directly to the reaction as a whole: the rate of production of iodine was assumed proportional to $[I^-]^2$, i.e. quadratic in iodide concentration. When the asymptotic analysis is conducted, the resulting model for switchover time (p6) is

$$t_{sw} \propto m_0^{-2},$$

which was found to fit the data well.

Reviewer 1's initial report highlighted an issue we had missed: several papers have been published which find that the rate of this reaction is proportional to $[I^-]$, i.e. it is linear in iodide concentration. The underlying mechanism is that the reaction involves multiple steps, with the rate limiting step being a nucleophilic attack of $[I^-]$ on H_2O_2 , as described in the reference [Copper & Koubek, J. Chem. Edu., 75(1):87-89, 1998] which the referee identified.

We therefore re-derived the entire mathematical model and asymptotic analysis with the linear law recommended by Reviewer 1, in an open-minded attempt to understand this issue. After all, we want to present the most accurate model of the system we are working with, and have no reason to prefer one model over another. Following the analysis through we found that with linear kinetics, the model for switchover time would take the form

$$t_{sw} \propto m_0^{-1}.$$

We then repeated the fitting procedure with this model (in fact fully expecting that Reviewer 1's suggestion would be correct). However, the new model could not fit the experimental data. Moreover, on a log-log plot, the experimental data followed the relationship

$$t_{sw} \propto m_0^{-2.3},$$

much closer to the original model than the one following from a linear law.

Below we give the best model fits (fitting two parameters for two experimental series simultaneously), which again strongly support the inverse-square model over the inverse-linear:

(a) original (quadratic) model

(b) referee's suggestion (linear) model

(The reason that the bottom left fit is so poor is that we are fitting both experimental series simultaneously: varying m_0 and varying c_0 , a more rigorous test than just fitting series independently).

While we were very ready to accept Reviewer 1's suggestion, supported by the references cited, the data were not consistent. On the other hand, the original quadratic model fits the data well. Our best explanation (given in our previous response to Reviewer 1) is that in our experiments there is a great excess of hydrogen peroxide in our experiment (0.09 mol/l as compared with 0.001–0.006 mol/l vitamin C – as now included in section 5), whereas for example Copper and Koubek worked with concentrations of iodide and peroxide that were quite similar. This excess of hydrogen peroxide ensures that the rate-limiting step (nucleophilic attack of I^- upon the H_2O_2 in equation (4) of Copper and Koubek, 1998) responsible for the linear rate law is no longer rate-limiting.

If the experimental data had supported the model suggested by Reviewer 1 we would have very willingly rewritten the paper around this, and indeed had carried out all the necessary new calculations. However, the data supported the original model.

Rebuttal to Reviewer 1's final response

Referee 1's final response did not engage with our detailed analysis of the consequences of a linear reaction law nor our new analysis of the experimental data, but rather reiterated their original comment, adding that "if it is the case [that H_2O_2 is in large excess], the H_2O_2 concentration does not change and can be included in the rate constant. This will turn the rate law in buffered media to rate = $k_0[I^-]$ ".

The latter claim was not evidenced by Reviewer 1; indeed we would reason that there must be a point at which further increases in H_2O_2 concentration simply saturate, so we do not understand the comment that the H_2O_2 concentration "can be included in the rate constant". It seems to us to be more probable that, because the overall reaction ultimately requires two I^- ions to come together, once the rate-limiting step is no longer rate-limiting, the overall reaction is governed by the quadratic kinetics expected from the law of mass action. In any case, we feel the issue is best decided by the fact that the data support the quadratic model, not the linear one.

We therefore appeal for the manuscript to be re-assessed on the basis that:

1. We assessed Reviewer 1's suggested alternative model on an equal footing with the original, and found that only the original model is consistent with the data.
2. Reviewer 1 did not respond to the fact that we had tested against experimental data, but rather focused on repeating the references of the first report (which are not necessarily relevant at high peroxide concentration), and making an un-evidenced suggestion that the linear law would still apply at high peroxide concentration – despite the fact that the data support the quadratic law.

Finally, we emphasise that, apart from the above issue, all three reviewers made very constructive and valuable comments, and regardless of what the outcome may be we thank them all for improving our work.

Appendix D

The manuscript 'Mathematical modelling of the vitamin C clock reaction' by Kerr, Thomson, and Smith, RSOS-181367.R1, now in a second version, is a very interesting study on the mathematical modeling of chemical reactions, specially for clock reactions. The study is deep and explore several aspects of the mathematical modeling of chemical reactions. The mathematical concepts looks correct. However, my main concern was not solved in this second version of the manuscript. As have indicated before, the chemical reaction

has a well established rate law*

$$\text{rate} = k[\text{I}^-][\text{H}_2\text{O}_2] \quad (\text{in buffered media})$$

$$\text{rate} = k[\text{I}^-][\text{H}_2\text{O}_2] + k'[\text{I}^-][\text{H}_2\text{O}_2][\text{H}^+] \quad (\text{considering the acid catalysis})$$

However, the authors consider the rate law indicated by eq (1.2)

$$\text{rate} = k_0[\text{I}^-]^2$$

arguing that H_2O_2 is in large excess. If it is the case, the H_2O_2 concentration does not change and can be included in the rate constant. This will turn the rate law in buffered media to

$$\text{rate} = k_0[\text{I}^-]$$

and not in the eq (1.2), which makes the mathematical work on this reaction unrealistic.

*References for the rate law of reaction (1.1), which were included in this new version of the manuscript:

1. H. A. Liebhasfky and A. Mohammad. The Kinetics of the Reduction, in Acid Solution, of Hydrogen Peroxide by Iodide Ion. *J. Am. Chem. Soc.*, 55(10):3977-3986, 1933.
2. C. L Copper and E. Koubek. A Kinetics Experiment To Demonstrate the Role of a Catalyst in a Chemical Reaction: A Versatile Exercise for General or Physical Chemistry Students. *J. Chem. Educ.*, 75(1):87-89, 1998.
3. P. D. Sattsangi. A Microscale Approach to Chemical Kinetics in the General Chemistry Laboratory: The Potassium Iodide Hydrogen Peroxide Iodine-Clock Reaction. *J. Chem. Educ.*, 88(2):184-188, 2011.